# Design and Synthesis of C-1 Methoxycarbonyl Derivative of Narciclasine and Its Biological Activity

**DOI:** 10.3390/molecules27123809

**Published:** 2022-06-14

**Authors:** Lihi Habaz, Korey Bedard, Mitchell Smith, Liqin Du, Alexander Kornienko, Tomas Hudlicky

**Affiliations:** 1Department of Chemistry, Brock University, 1812 Sir Isaac Brock Way, St. Catharines, ON L2S 3A1, Canada; kbedard@brocku.ca; 2Department of Chemistry and Biochemistry, Texas State University, San Marcos, TX 78666, USA; mts85@txstate.edu (M.S.); l_d141@txstate.edu (L.D.)

**Keywords:** narciclasine, chemoenzymatic, synthesis, enantioselective synthesis, natural products, toluene dioxygenase

## Abstract

A 15-step chemoenzymatic total synthesis of C-1 methoxycarbonyl narciclasine (**10**) was accomplished. The synthesis began with the toluene dioxygenase-mediated dihydroxylation of *ortho*-dibromobenzene to provide the corresponding *cis*-dihydrodiol (**12**) as a single enantiomer. Further key steps included a nitroso Diels–Alder reaction and an intramolecular Heck cyclization. The C-1 homolog **10** was tested and evaluated for antiproliferative activity against natural narciclasine (**1**) as the positive control. Experimental and spectral data are reported for all novel compounds.

## 1. Introduction

Narciclasine (**1**) and pancratistatin (**2**), Figure 1, are two alkaloids found in the Amaryllidaceae family of plants that have garnered significant scientific attention due to their attractive biological activities. The antineoplastic properties of (**1**) and (**2**) have led to intense investigations towards their synthesis, and their scarce availability and poor solubility profiles have also led to synthetic efforts related to their unnatural and/or truncated analogs. Thus, a common objective of these endeavors is to synthesize products with comparable or enhanced anti-tumor activities to those of (**1**) and (**2**) with improved bioavailability profiles. In this paper we present our most recent efforts to synthesize such novel compounds, namely the C-1 methoxycarbonyl analog of narciclasine (**10**), and report findings on its biological activity.

Previous studies [1] of the pharmacophore of Amaryllidaceae alkaloids suggest that structural alterations in the “northwest bay region”, encompassing the region between C-1 and C-10, can be introduced without detriment to the biological activities. The C-1 benzoate derivative of pancratistatin (**3**) reported by Pettit [2] and the C-1 benzoyloxymethyl compound (**4**) reported by us [3,4] are the only synthetic derivatives of pancratistatin that exhibited enhanced biological activity when tested (both exhibiting nanomolar activity) [5,6,7,8,9]. These findings suggest that further studies of both pancratistatin and narciclasine analogs may lead to the discovery of novel compounds that exhibit similar or enhanced activities. Pancratistatin is more thoroughly discussed in the literature, even though narciclasine was shown to have the stronger biological activity [10]. The number of unnatural derivatives of narciclasine reported in the literature is very small compared with those derived from pancratistatin. Thus, our group previously published efforts towards the total syntheses of narciclasine analogs, namely compounds **5**–**9** shown in Figure 1 [11,12,13,14]. 10-Aza-narciclasine (**7**) is the only derivative of narciclasine synthesized by our group that displayed activity comparable to that of the natural product [12].

Recently, we began pursuing the synthesis of C-1 narciclasine analogs based on a convergent synthesis wherein the A- and C-ring precursors are synthesized separately and later combined to form the B-ring, thus producing the full narciclasine backbone (Figure 1). The synthesis of the A-ring precursor (**14**) begins from *ortho*-vanillin (**15**) and is based on previous work [15,16,17,18], while the C-ring synthesis begins with *ortho*-dibromobenzene (**13**). Enzymatic dihydroxylation of the latter compound installs the *syn*-diol moiety at C-3 and C-4 in the correct configuration (**12**, Figure 1) [19]. Key steps in the synthesis include a nitroso Diels–Alder reaction that installs the heteroatoms at C-2 and C-5 in the correct stereochemical configuration, and an intramolecular Heck cyclization that forms the C10a-C10b bond. In this paper we report the first total synthesis of a C-1 narciclasine homolog, as well as initial biological evaluation studies for the synthesized compound.

## 2. Results and Discussion

### 2.1. Synthesis of the A-Ring Coupling Precursor

The synthesis of the A-ring coupling precursor of narciclasine is based on previous Amaryllidaceae chemistry [15,16] that involves the derivatization of *ortho*-vanillin (**15**) to the bromoarene (**18**) in three steps (Figure 2). This is followed by a Rieche formylation reaction [17] that installs an aldehyde moiety at C-6a (**19**) that can be further elaborated to produce the acid chloride **14** in two additional steps [18]. At this stage, the crude acid chloride **14** is coupled to the primary amine of the C-ring precursor (**11**) to form the necessary amide bond of the B-ring. 

### 2.2. Enzymatic Dihydroxylation of ortho-Dibromobenzene and Nitroso Diels–Alder Reaction

The more elaborate synthesis of the C-ring coupling precursor, which contains all the necessary chiral centers of the final product, begins with the enzymatic dihydroxylation of *ortho*-dibromobenzene (**13**, Figure 3). This is accomplished by a whole-cell fermentation process [20] with a recombinant strain of *E. coli* JM109 (pDTG601A [21]) that over-expresses toluene dioxygenase and was successfully used for the oxidative de-aromatization of various aromatic substrates [22,23,24,25,26,27,28,29,30,31,32,33,34,35,36,37] employed in the synthesis of natural products. Once dibromodiene diol **12** was obtained, it was subjected to acetonide protection and a nitroso Diels–Alder reaction in a one-pot procedure to form the bicyclic oxazine **21**. The facial selectivity that is observed in the produced oxazine is attributed to the presence of the acetonide group at C-3 and C-4 that effectively blocks the dienophile approach from the β-face. Additionally, the observed regioselectivity of the cycloaddition reaction stems from the C-Br bond at C-2 that results in a highly polarized diene. 

### 2.3. C–1 Functionalization via Pd-Catalyzed Carbonylation

Upon reduction of the N-O bond of the nitroso Diels–Alder adduct **21** (Figure 4), conduramine **22** is obtained which contains all the necessary carbon–heteroatom bonds of the C-ring in the correct stereochemical configurations. This operation was performed by treating **21** with aluminum amalgam which simultaneously reduced the N-O bond as well as the C-Br bond at C-2. The product is obtained in modest yields, due to the overreduction of the C-Br bond at C-1 as an unavoidable competing reaction. 

After the TBS protection of the C-2 hydroxyl group, the next step is functionalization at C-1. Initial studies concentrated on the conversion of the vinyl bromide moiety of **23** to a vinyl nitrile moiety (**25**, Figure 5), with the end-goal being reduction of the nitrile to an aldehyde group. This was done because installation of an aldehyde at C-1 has the potential for further derivatization to both hydroxymethyl and aminomethyl derivatives. While the installation of a nitrile group at C-1 was performed successfully, treatment of the resulting nitrile **25** with a hydride source (for example, DIBAL-H) resulted in the reduction of the nitrile with concomitant loss of the C-2 silyl ether group, presumably via a 1,4-conjugate elimination reaction (Figure 5). Thus, a different route was pursued based on previous work [38] from our group in which a Pd-catalyzed carbonylation reaction was carried out in the presence of carbon monoxide and methanol to produce the corresponding methyl ester (**24**, Figure 4). 

The final operation that was performed on this coupling precursor is the deprotection of the benzyl carbamate to reveal a primary amine (**11**, Figure 4) that can be condensed with the A-ring coupling partner, acid chloride **14**. This could not be carried out under classical hydrogenolysis [39] or hydrolysis conditions because of the presence of incompatible functional groups, namely the C10b-C1 olefin and the C-1 methyl ester. However, we were pleased to learn that this transformation can be successfully carried out by subjecting the substrate to a transfer hydrogenation reaction [40] by treatment of **24** with triethylsilane and palladium chloride to furnish amine **11**. 

### 2.4. Intramolecular Heck Reaction and Global Deprotection to Afford 1-Methoxycarbonyl Narciclasine

Following the deprotection of the benzyl carbamate, the resulting amine (**11**) was subjected to a condensation reaction with acid chloride **14** to produce adduct **27** (Figure 6) which contains the full narciclasine backbone, except for the C10a–C10b bond. Following a Boc protection reaction to form imide **28**, the key intramolecular Heck reaction was performed to establish this bond. This transformation was successfully accomplished to afford the cyclized product (**29**) by employing reaction conditions that were previously used by our group in the synthesis of aza-narciclasine derivatives **5**, **6** and **7**. The final stage of this synthesis is the global deprotection of **29** to furnish the desired product in two more steps. We again referred to previously established protocols that our group used in the synthesis of pancratistatin and narciclasine analogs [11,12]. Demethylation was carried out first (LiCl, DMF, 90 °C) and under these conditions thermal *N*-Boc deprotection also occurred to obtain **30**. Desilylation and acetonide deprotection were successfully accomplished by treating **30** with HCl to obtain the target product, 1–methoxycarbonyl narciclasine (**10**). 

### 2.5. Biological Activity

We used an in vitro cancer model—lung adenocarcinoma cell line A549 (Figure 2)—to evaluate the ability of compound **10** to inhibit the proliferation of cancer cells. Narciclasine was used as a control. As expected, the narciclasine control showed good nanomolar activity against this cell line, displaying an IC_50_ of 20 nM. Compound **10** was much less potent, yielding an IC_50_ of 15.5 μM (Figure 2). Thus, it appears that derivatization of narciclasine at C-1 with the methoxycarbonyl group leads to a reduction in activity although it does not abolish it completely.

## 3. Materials and Methods

### 3.1. General Information

All solvents were distilled and kept dry before usage. All reactions were done in inert atmosphere (Ar or N_2_) and at room temperature, unless otherwise stated. All reagents were obtained from commercial sources. Nuclear magnetic resonance (NMR) analyses were performed on Bruker Avance AV 300, Bruker Avance III HD 400 and Bruker Avance AV 600 digital NMR spectrometers, running Topspin 2.1 and 3.5 software. The probes are furnished with VT (variable temperature) and gradient equipment. Chemical shifts are given in δ, relative integral, multiplicity (singlet (s), doublet (d), triplet (t), quartet (q), multiplet (m)) and coupling constants (*J*) in Hz. Melting points (m.p.) were measured in a capillary apparatus. Mass spectra (HRMS) measurements were determined on an LTQ Orbitrap XL. The molecular mass-associated ion was measured by electron ionization, electrospray ionization or fast atom bombardment. Infrared (IR) spectra were recorded on an FT-IR spectrophotometer as neat and are reported in wave numbers (cm^−1^) and intensity (broad (br), strong (s), medium (m), weak (w)). Column chromatography was performed on flash grade 60 silica gel. Thin-layer chromatography (TLC) was performed on silica gel 60 F254-coated aluminum sheets. TLC plates were visualized using UV and stained with iodine, cerium ammonium molybdate (CAM), KMnO_4_ solutions, FeCl_3_ solutions, ninhydrin solutions or 2,4-dinitrophenylhydrazine (2,4-DNP) solutions.

The five-step synthesis of compound **20** from *ortho*-vanillin (**15**) was accomplished by repetition of known methods [15,16,17,18]. Experimental and spectral data are reported for all novel compounds (see Appendix A).

### 3.2. Preparation and Characterization of Compounds

#### 3.2.1. (3a*S*,4*S*,7*R*,7a*S*)-Benzyl-4,5-dibromo-2,2-dimethyl-3a,4,7,7a-tetrahydro-4,7-(epoxyi-mino)benzo[d][1,3]dioxole-8-carboxylate (**21**)

To the solution of dibromo dienediol **12** [19,20] (1.68 g, 6.2 mmol) in 2,2-DMP (40 mL) was added *p*TsOH (107 mg, 0.6 mmol). This mixture was allowed to stir at room temperature for approximately 2 h until a TLC analysis showed complete consumption of the starting material. At this point, NaIO_4_ (2.65 g, 12.4 mmol) and H_2_O (10 mL) were added to the reaction mixture and the contents were cooled to 0 °C. A separate solution of benzyl hydroxycarbamate (CbzNHOH, 1.55 g, 9.3 mmol) in MeOH (20 mL) was prepared and added dropwise to the reaction mixture over a period of 30 min. The reaction mixture was left to warm up to room temperature and stirred for 16 h at room temperature, then it was quenched with a saturated NaHSO_3_ solution (20 mL) and the volume of the mixture was reduced by rotary evaporation. The aqueous layer was extracted with EtOAc (3 × 40 mL), the organic layers were combined and dried with Na_2_SO_4_, filtered, and the filtrate was concentrated by rotary evaporation. The crude product was purified by column chromatography [silica gel, EtOAc:Hex (1:3)] to obtain **21** as a colorless gel (1.9 g, 74 %).

**21**: R*_f_* = 0.6 [EtOAc:Hex (1:3)]; ${\left[\alpha \right]}_{D}^{23}$ = −5.4 (*c* = 0.36, CHCl_3_); IR (neat, cm^−1^) 3030, 2997, 1755, 1725, 1598, 1442, 1377, 1268, 1225, 1074; ^1^H NMR (300 MHz, CDCl_3_) δ 7.39–7.30 (m, 5H), 6.69 (d, *J* = 6.1 Hz, 1H), 5.21 (s, 2H), 5.08 (dd, *J* = 6.4, 4.1 Hz, 1H), 4.79 (d, *J* = 6.9 Hz, 1H), 4.56 (dd, *J* = 6.9, 4.0 Hz, 1H), 1.39 (s, 3H), 1.33 (s, 3H); ^13^C NMR (75 MHz, CDCl_3_) δ 157.3, 135.2, 131.3, 128.6 (2C), 128.5, 127.9 (2C), 122.1, 111.9, 93.8, 82.3, 73.8, 68.7, 55.4, 25.8, 25.5. HRMS (EI) calcd for C_16_H_14_Br_2_NO_5_[M-CH_3_]^+^: 457.9233. Found 457.9232; Anal. Calcd for C_17_H_17_Br_2_NO_5_: C, 42.97; H, 3.61. Found C, 43.19; H, 3.85.

#### 3.2.2. ((3a*S*,4*R*,7*R*,7a*R*)-Benzyl-6-bromo-7-hydroxy-2,2-dimethyl-3a,4,7,7a-tetrahydroben-zo[*d*][1,3]dioxol-4-yl)carbamate (**22**)

Aluminum amalgam (prepared from 20 mg, 0.72 mmol, 8 equivalents of aluminum turnings after treatment with 1 M aq. KOH and 0.5% aq. HgCl_2_) was added to a stirred solution of the oxazine **21** (43 mg, 0.09 mmol) in aqueous THF (THF:H_2_O, 10:1, 1 mL), cooled to 0 °C, over 3 days (2 equivalents at reaction start time, then another 2 equivalents every 24 h). After each addition, the reaction was allowed to attain room temperature and stirring was continued until the next addition. After about 96 h, TLC analysis confirmed reaction completion. The reaction mixture was diluted with 2 mL of MeOH, stirred for 10 min, then filtered through Celite. The organic layer was dried with Na_2_SO_4_ and concentrated. The residue was purified by column chromatography [silica gel, EtOAc:Hex (1:2)] to afford the alcohol **22** as a colorless oil (23 mg, 40%).

**22:** R*_f_* = 0.5 [EtOAc:Hex (1:2)]; ${\left[\alpha \right]}_{D}^{23}$ = −20.7 (*c* = 0.46, CHCl_3_); IR (neat, cm^−1^) 3334, 2933, 1697, 1516, 1374, 1240, 1212, 1035; ^1^H NMR (300 MHz, CDCl_3_) δ 7.34 (s, 5H), 6.33 (d, J = 6.1 Hz, 1H), 5.49 (d, J = 9.0 Hz, 1H), 5.09 (s, 2H), 4.50 (dd, J = 6.7, 2.5 Hz, 1H), 4.43–4.30 (m, 3H), 3.41 (s, 1H), 1.40 (s, 3H), 1.31 (s, 3H); ^13^C NMR (75 MHz, CDCl_3_) δ 155.8, 136.2, 130.3, 128.7, 128.4 (2C), 128.3 (2C), 126.9, 108.9, 79.0, 75.9, 73.1, 67.3, 50.9, 26.7, 24.6. HRMS (EI) calcd for C_16_H_17_BrNO_5_[M-CH_3_]^+^: 382.0285. Found 382.0280; Anal. Calcd for C_17_H_20_BrNO_5_: C, 51.27; H, 5.06. Found C, 51.50; H, 5.11.

#### 3.2.3. ((3a*S*,4*R*,7*R*,7a*S*)-Benzyl-6-bromo-7-((*tert*-butyldimethylsilyl)oxy)-2,2-dimethyl-3a,4,7,7a-tetrahydrobenzo[*d*][1,3]dioxol-4-yl)carbamate (**23**)

Imidazole (136 mg, 2.0 mmol) and TBSCl (452 mg, 3.0 mmol) were added to a solution of alcohol **22** (402 mg, 1.0 mmol) in 10 mL of DCM at room temperature. After the complete consumption of the starting material was observed by TLC (approximately 24 h), the reaction mixture was quenched with a saturated aqueous solution of NH_4_Cl (10 mL) and extracted with DCM (3 × 10 mL). The combined organic phases were dried over Na_2_SO_4_ and concentrated. The residual oil was purified by column chromatography [silica gel, EtOAc:Hex (1:4)] to afford the product **23** as a pale-yellow oil (439 mg, 86%). 

**23:** R*_f_* = 0.6 [EtOAc:Hex (1:4)]; ${\left[\alpha \right]}_{D}^{22}$ = −4.5 (*c* = 0.30, CHCl_3_); IR (neat, cm^−1^) 3398, 2931, 1720, 1637, 1512, 1384, 1224, 1065; ^1^H NMR (300 MHz, CDCl_3_) δ 7.35 (s, 5H), δ 6.50 (d, *J* = 6.5 Hz, 1H), δ 5.69 (d, *J* = 5.69 Hz, 1H), δ 5.11 (s, 1H), δ 4.55 (s, 1H), δ 4.36 (s, 1H), δ 1.40 (s, 3H), δ 1.32 (s, 3H), δ 0.91 (s, 9H), δ 0.22 (s, 6H); ^13^C NMR (75 MHz, CDCl_3_) δ 155.7, 136.6, 131.6, 128.7, 128.3, 127.4, 109.0, 80.1, 74.7, 67.1, 49.6, 30.0, 26.6, 25.9, 24.8, 18.1, −4.1, −4.8. HRMS (EI) calcd for C_22_H_31_BrNO_5_Si[M-CH_3_]^+^: 496.1149. Found 496.1149. Anal. Calcd for C_23_H_34_BrNO_5_Si: C, 53.90; H, 6.69. Found C, 54.00; H, 6.89.

#### 3.2.4. (3a*S*,4*S*,7*R*,7a*S*)-Methyl 7-(((benzyloxy)carbonyl)amino)-4-((*tert*-butyldimethylsilyl) oxy)-2,2-dimethyl-3a,4,7,7a-tetrahydrobenzo[*d*][1,3]dioxole-5-carboxylate (**24**)

Pd(PPh_3_)_4_ (81 mg, 0.07 mmol) and distilled Et_3_N (0.08 mL, 0.54 mmol) were added to an anhydrous stirred solution of the vinyl bromide **23** (137 mg, 0.27 mmol) in DMF:MeOH (1:1, 4 mL). The mixture was degassed with CO gas for 15 min and then a CO balloon was fixed to the top of the flask and it was heated to 65 °C for 24 h. Once complete consumption of the starting material was observed by TLC analysis, the reaction mixture was allowed to cool to room temperature and then diluted with DCM and filtered through a plug of Celite. The resulting filtrate was extracted with a saturated solution of NH_4_Cl (5.0 mL) and washed with DCM (3 × 10 mL). The combined organic layers were dried over Na_2_SO_4_ and concentrated. The residue was purified by column chromatography [silica gel, EtOAc:Hex (1:4)] to obtain the product **24** as a white solid (72 mg, 54%).

**24:** R*_f_* = 0.5 [silica gel, EtOAc:Hex (1:4)]; mp 77 °C (EtOAc); ${\left[\alpha \right]}_{D}^{22}$ = −12.3 (*c* = 0.10, CHCl_3_); IR (neat, cm^−1^) 3383, 3066, 3033, 2952, 2930, 2900, 2857, 1717, 1504; ^1^H NMR (300 MHz, CDCl_3_) δ 7.39–7.27 (m, 5H), 5.77 (d, *J* = 10.3 Hz, 1H), 5.14–5.01 (m, 2H), 4.79 (s, 1H), 4.75–4.64 (m, 1H), 4.48 (dt, *J* = 6.8, 4.3 Hz, 2H), 3.79 (s, 3H), 1.28 (d, *J* = 2.5 Hz, 3H), 1.27 (s, 3H), 0.83 (s, 9H), 0.18 (s, 3H), 0.06 (s, 3H). ^13^C NMR (100 MHz, CDCl_3_) δ 166.1, 155.8, 140.9, 136.5, 135.6, 128.7, 128.3, 108.6, 88.0, 78.6, 77.8, 77.4, 76.9, 72.7, 67.2, 64.9, 52.4, 47.2, 26.4, 26.0, 24.4, 18.2, −4.5, −4.9. HRMS (EI) calcd for C_25_H_37_NO_7_Si: 491.2334. Found 491.2338. 

#### 3.2.5. (3a*S*,4*S*,7*R*,7a*S*)-Methyl-7-(6-bromo-4-methoxybenzo[*d*][1,3]dioxole-5-carboxamido) -4-((*tert*-butyldimethylsilyl)oxy)-2,2-dimethyl-3a,4,7,7a-tetrahydrobenzo[*d*][1,3]dioxole-5 -carboxylate (**27**)


*Step A: Preparation of acid chloride* **14**


A flame-dried flask equipped with a stir bar was charged with the acid **20** (509 mg, 1.85 mmol). After the flask contents were dissolved in DCM (16 mL), (COCl)_2_ was added (0.32 mL, 3.7 mmol). A catalytic amount of DMF was then added (approximately 5 drops) to the solution, and the reaction mixture was allowed to stir at room temperature for 3 h. The volume of the reaction mixture was then reduced under a rotary evaporator [which was equipped with a bubbler containing a 10% NaOH (aq.) solution] and the crude acid chloride **14** was then dried further under high vacuum. It was immediately used in step C without further purification.


*Step B: Preparation of primary amine* **11**


A flame-dried flask equipped with a stir bar was charged with the substrate **24** (455 mg, 0.93 mmol) and PdCl_2_ (181 mg, 1.02 mmol). After the contents were dissolved in DCM (22 mL), Et_3_N (0.39 mL, 2.8 mmol) and Et_3_SiH (0.45 mL, 2.8 mmol) were added, respectively. After TLC analysis confirmed the complete consumption of the starting material (30 min), the reaction mixture was filtered through a plug of Celite and concentrated to dryness under high vacuum. It was immediately used in step C without further purification. 



*Step C: Condensation of amine **11** with acid chloride **14** to afford compound **27***



The crude amine **11** (0.93 mmol) and DMAP (11 mg, 0.09 mmol) were charged to flame-dried flask equipped with a stir bar. The reaction mixture contents were then dissolved in DCM (14.0 mL) and pyridine (0.23 mL, 2.8 mmol) was added. The reaction mixture was then cooled to 0 °C. A separate solution of the crude acid chloride **14** (1.85 mmol) in DCM (8.0 mL) was prepared and added to the reaction mixture over 10 min. The reaction mixture was then allowed to slowly warm to room temperature over 1 h and then stirred for another 12 h. Next, the reaction mixture was concentrated and dried under high vacuum, and the residue purified by column chromatography [silica gel, EtOAc:Hex (1:1)] to obtain the product **27** as a pale-yellow oil (411 mg, 72%).

**27**: R*_f_* = 0.6 [silica gel, EtOAc:Hex (1:1)]; ${\left[\alpha \right]}_{D}^{22}$ = −28.0 (*c* = 0.30, CHCl_3_). IR (neat, cm^−1^) 3369, 2927, 2855, 1721, 1679,1504, 1468; ^1^H NMR (300 MHz, CDCl_3_) δ 6.70 (s, 1H), 6.64 (d, *J* = 10.3 Hz, 1H), 6.02 (s, 1H), 5.96 (s, 2H), 5.26 (dd, *J* = 14.6, 4.5 Hz, 1H), 4.83 (s, 1H), 4.66 (d, *J* = 6.9 Hz, 1H), 4.51 (d, *J* = 4.7 Hz, 1H), 3.99 (s, 3H), 3.82 (s, 3H), 1.33 (s, 3H), 1.31 (s, 3H), 0.63 (s, 7H), 0.14 (s, 3H), 0.01 (s, 3H). ^13^C NMR (150 MHz, CDCl_3_) δ 165.9, 164.8, 150.4, 141.2, 140.4, 136.0, 135.8, 125.5, 111.3, 108.5, 107.2, 101.8, 78.4, 76.5, 64.9, 60.4, 52.2, 45.4, 26.2, 25.3, 24.2, 17.5, −4.9, −5.0. HRMS (EI) Calcd for C_26_H_36_BrNO_9_Si: 613.1337, Found 613.1328. 

#### 3.2.6. (3a*S*,4*S*,7*R*,7a*S*)-Methyl-7-(6-bromo-*N*-(*tert*-butoxycarbonyl)-4-methoxybenzo[*d*][1,3] dioxole-5-carboxamido)-4-((*tert*-butyldimethylsilyl)oxy)-2,2-dimethyl-3a,4,7,7a-tetrahyd-robenzo[*d*][1,3]dioxole-5-carboxylate (**28**)

The amide **27** (388 mg, 0.63 mmol) and DMAP (158 mg, 1.29 mmol) were charged to a flame-dried RBF with a stir bar, then dissolved in dry MeCN (12 mL). Next, Boc_2_O (275 mg, 1.26 mmol) was added. The reaction was left to stir at room temperature for 4 h before quenching with H_2_O (20 mL) and extracting with DCM (3 × 20 mL). The organic layers were then combined, dried with MgSO_4_ and concentrated to dryness under reduced pressure. The residue was purified by column chromatography [silica gel, EtOAc:Hex (1:2)] to obtain **28** (338 mg, 75%) as a yellow oil. 

**28**: R*_f_* = 0.6 [silica gel, EtOAc:Hex (1:2)]. ${\left[\alpha \right]}_{D}^{22}$ = −60.4 (*c* = 0.45, CHCl_3_). IR (neat, cm^−1^) 2986, 2956, 2934, 2859, 1729, 1679, 1623. ^1^H NMR (mixture of rotamers) (300 MHz, CDCl_3_) δ 6.72 (d, *J* = 3 Hz, 1H), 6.62 (s, 1H), 5.97 (s, 2H), 5.52 (s, 1H), 4.78 (s, 1H), 4.57 (m, 1H), 4.30 (m, 1H), 3.97 (s, 3H), 3.74 (s, 3H), 1.52 (m, 3H), 1.39 (m, 3H), 1.33 (m, 9H), 0.87 (s, 9H), 0.20 (s, 3H), 0.13 (s, 3H). ^13^C NMR (150 MHz, CDCl_3_) δ 166.5, 151.5, 150.1, 148.0, 137.7, 130.7, 108.6, 107.4, 107.1, 102.0, 84.4, 82.2, 80.3, 76.7, 66.2, 64.0, 62.5, 60.4, 60.3, 54.5, 51.8, 27.8, 26.4, 25.9, 18.1, −4.6, −5.0. HRMS (EI) Calcd for C_31_H_44_BrNO_11_Si: 713.1862, Found 713.1878.

#### 3.2.7. 4-(*tert*-Butyl)-11-methyl(3a*S*,3b*R*,12*S*,12a*S*)-12-((*tert*-butyldimethylsilyl)oxy)-6-met-hoxy-2,2-dimethyl-5-oxo-3a,5,12,12a-tetrahydrobis([1,3]dioxolo)[4,5-c:4’,5’-j]phenanthrid-ine-4,11(3bH)-dicarboxylate (**29**)

A flame-dried flask with a stir bar was charged with the substrate **28** (123 mg, 0.17 mmol), Pd(OAc)_2_ (10 mg, 0.04 mmol) and Ag_3_PO_4_ (214 mg, 0.51 mmol). The reaction mixture contents were then dissolved in dry toluene (9 mL) and degassed with argon for 15 min. Next, 1,2–*bis*(diphenylphosphino)ethane (14 mg, 0.03 mmol) was added and the reaction mixture was heated to 130 °C for 16 h. After cooling to room temperature, the reaction mixture was filtered through a plug of Celite and then concentrated under reduced pressure. The residue was purified by column chromatography [silica gel, EtOAc:Hex (1:2)] to obtain **29** (44 mg, 40%, 61% brsm) as a colorless oil and 41 mg of recovered starting material **28**. 

**29**: R*_f_* = 0.4 [silica gel, EtOAc:Hex (1:2)]. ${\left[\alpha \right]}_{D}^{22}$ = 130.0 (*c* = 0.8, CHCl_3_). IR (neat, cm^−1^) 2989, 2956, 2934, 2857, 1722, 1666, 1606. ^1^H NMR (400 MHz, CDCl_3_) δ 6.71 (s, 1H), 6.03 (d, *J* = 1.4 Hz, 2H), 4.89 (d, J = 6.1 Hz, 1H), 4.58–4.54 (m, 1H), 4.08–4.04 (m, 2H), 4.03 (s, 3H), 3.76 (s, 3H), 1.53 (s, 3H), 1.52 (s, 9H), 1.31 (s, 3H), 0.90 (s, 9H), 0.17 (s, 3H), 0.14 (s, 3H). ^13^C NMR (100 MHz, CDCl_3_) δ 168.1, 161.4, 152.7, 152.1, 144.4, 139.6, 133.6, 129.7, 126.6, 118.3, 111.5, 102.4, 100.2, 83.1, 79.4, 79.0, 73.4, 61.4, 56.0, 52.5, 28.1(3C), 27.2, 25.8 (3), 25.1, 18.2, −4.3, −5.1. HRMS (EI) calcd for C_27_H_34_NO_11_Si [M-C_4_H_9_]^+^: 576.1896. Found 576.1905.

#### 3.2.8. (3a*S*,3b*R*,12*S*,12a*S*)-Methyl-12-((*tert*-butyldimethylsilyl)oxy)-6-hydroxy-2,2-dimet-hyl-5-oxo-3a,3b,4,5,12,12a-hexahydrobis([1,3]dioxolo)[4,5-c:4’,5’-j]phenanthridine-11-car-boxylate (**30**)

LiCl (28 mg, 0.66 mmol) was added to a solution of the substrate **29** (28 mg, 0.044 mmol) in dry DMF (3 mL). The reaction mixture contents were then subjected to three cycles of freeze-pump-thaw, before the mixture was heated to 90 °C for 12 h. The reaction mixture was then cooled to room temperature and filtered through a plug of Celite, then concentrated under reduced pressure. The residue was purified by column chromatography [silica gel, EtOAc:Hex (1:2)] to obtain **30** (18 mg, 80%) as a white solid. 

**30**: R*_f_* = 0.6 [silica gel, EtOAc:Hex (1:2)]; mp 260 °C (decomp., EtOAc); ${\left[\alpha \right]}_{D}^{22}$ = 38.4 (*c* = 0.85, CHCl_3_). IR (neat, cm^−1^) 3205, 3181, 3088, 3034, 2997, 2954, 2929, 2857, 1731, 1672, 1620, 1599. ^1^H NMR (600 MHz, CDCl_3_) δ 13.06 (s, 1H), 6.57 (s, 1H), 6.31 (s, 1H), 6.05 (s, 2H), 4.49 (d, *J* = 1 Hz, 1H), 4.17 (dd, *J* = 2, 1 Hz, 1H), 4.08 (dd, *J* = 2, 1 Hz, 1H), 4.03 (d, *J* = 1 Hz, 1H), 3.82 (s, 3H), 1.51 (s, 3H), 1.36 (s, 3H), 0.91 (s, 9H), 0.18 (s, 3H), 0.14 (s, 3H). ^13^C NMR (150 MHz, CDCl_3_) δ 168.4, 167.4, 152.9, 146.7, 135.4, 132.2, 126.9, 124.2, 111.5, 106.2, 102.7, 97.5, 79.1, 78.2, 73.5, 54.9, 52.7, 27.3, 25.8, 25.1, 18.1, −4.3, −5.2. HRMS (EI) calcd for C_25_H_33_NO_9_Si: 519.1919. Found 519.1923.

#### 3.2.9. (2*S*,3*R*,4*S*,4a*R*)-Methyl-2,3,4,7-tetrahydroxy-6-oxo-2,3,4,4a,5,6-hexahydro-[1,3]diox-olo [4,5-j]phenanthridine-1-carboxylate (**10**)

HCl (0.1 mL, 1.1M) was added to a solution of the substrate **30** (16 mg, 0.031 mmol) in THF (1 mL) that was cooled to 0 °C. The reaction mixture was allowed to attain room temperature over 1 h and then stirred at room temperature for a further 36 h. The reaction mixture was then extracted with EtOAc (5 mL × 5) and the combined organic layers were dried with MgSO_4,_ then concentrated under reduced pressure. The residue was purified by column chromatography [silica gel, MeOH:DCM (1:5)] to obtain **10** (6 mg, 55%) as a white solid.

**10**: R*_f_* = 0.6 [silica gel, MeOH:DCM (1:5)]; mp > 300 °C (decomp., EtOAc); ${\left[\alpha \right]}_{D}^{20}\;$ = 141.5 (*c* = 0.30, MeOH); IR [neat, cm^−1^] 3666, 3297, 3099, 2983, 2907, 2796, 2669, 2119, 1698, 1672, 1595, 1508; ^1^H NMR [400 MHz, (CD_3_)_2_CO] δ 12.91 (s, 1H), 7.39 (s, 1H), 6.29 (s, 1H), 6.09 (dd, J = 7.5, 1.1 Hz, 2H), 4.78 (s, 2H), 4.65 (d, J = 3.0 Hz, 1H), 4.42–4.35 (m, 1H), 4.05 (t, J = 3.0 Hz, 1H), 4.04 (s, 1H), 3.64 (s, 3H); ^13^C NMR [100 MHz, (CD_3_)_2_CO] δ 170.8, 169.9, 152.9, 146.5, 135.8, 132.6, 131.1, 130.3, 108.1, 103.5, 99.3, 73.7, 72.8, 70.3, 54.7, 52.4; HRMS (EI) calcd. for C_16_H_15_NO_9_: 365.0741. Found 365.0749.

### 3.3. Cell Viability Assay

Cell viability was measured by MTT (3-(4,5-dimethylthiazol-2-yl)-2,5-diphenyl tetrazolium bromide) assay in lung adenocarcinoma cell line A549 (obtained from American Type Culture Collection). A549 cells were maintained in RPMI-1640 media supplemented with 10% FBS. For measuring the IC_50_ values of cell viability, 3000 cells per well were plated into a 96-well plate, and the cells were treated with a series of dilutions (from 500 µM to 0.32 nM) of individual compounds in triplicate for four days. After four days, the cells were replaced with MTT reagent (at 0.25 mg/mL in culture media) in each well and incubated with cells for 2 h at 37 °C. The culture media was removed after the microplates were spun at 2000 rpm for 5 min. DMSO was then used to dissolve the crystals formed in each of the 96 wells. Optical density values at wavelength 570 nm and 630 nm were measured using SpectraMax 190 (Molecular Devices, San Jose, CA, USA), and the difference in the two optical density values was used to analyze the relative cell survival in each well. IC_50_ values were calculated using Graphpad Prism software.

## 4. Conclusions

The first C-1 homolog of narciclasine, methoxycarbonyl narciclasine (**10**), was prepared in fifteen steps via a chemoenzymatic total synthesis from *ortho*-dibromodiene diol (**12**) and *ortho*-vanillin (**15**). The enhanced biological activities of C-1 homologs of pancratistatin (such as compounds **3** and **4** [2,3]) provided the incentive to pursue the analogous synthesis of C-1 homologs of narciclasine. Our synthesis and biological testing of compounds **9**, described elsewhere, and **10**, described here, show that derivatization of narciclasine at position C-1 leads to reduced antiproliferative potencies, although it does not abolish activity completely. This stands in contrast to similar efforts in pancratistatin area where more potent analogs can be achieved through C-1 modification [2,3]. Current efforts are underway to synthesize other unnatural C-1 narciclasine homologs.

## Data Availability

Not applicable.

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
