# Peer review of "Design and Synthesis of C-1 Methoxycarbonyl Derivative of Narciclasine and Its Biological Activity"

_molecules, 2022, doi:10.3390/molecules27123809_

Round 1
Reviewer 1 Report
In this manuscript, the authors demonstrated the synthesis of 1-methoxycarbonyl narciclasine starting from o-vanillin and 1,2-dibromobenzene via a 15-step sequence. The second part of the synthetic route is novel (9 new compounds: 10, 21-24 and 27-30), however, the first part of the synthetic sequence can be found in the literature. The reactions were performed in moderate to good yields. The structures of the synthesized compounds have been adequately characterized by NMR spectroscopy and mass spectrometry. Unfortunately, the in vitro antitumor activity of the end-product against lung adenocarcinoma cell line A549 is approximately 3 orders of magnitude lower than that of control narciclasine, which is a well-known naturally occurring Amaryllidaceae alkaloid. In summary, I think that this manuscript may be suitable for publication in the Molecules after some modifications.
General remarks:
The title is too general and misleading because of the plurals. Suggestion: Design and Synthesis of C-1 Methoxycarbonyl Derivative of Narciclasine and Its in Vitro Anticancer Activity
A recurring mistake is that the name of the -COOMe group is not carboxymethyl but methoxycarbonyl. (The carboxymethyl group is -CH2COOH.) This should be corrected in the following lines: L10, L28, L191, in Scheme 6, L209, L224, L231 and L474.
1-Hydroxynarciclasine is not described in SciFinder. Do you have a reference for it? I didn't find it at ref [6]. It would be interesting to have some information about its effect, as the target product is also designed to be functionalized in position 1 of narciclasine.
The manuscript needs a revision:
L13: intramolecular instead of Intramolecular
32-33: „The C-1 benzoate derivative of pancratistatin (3)” instead of „The pancratistatin C-1 benzoate derivative (3)”
33-34: The name of the -CH2OBz group is not methylbenzoyl. Suggestions: „the C-1 benzoyloxymethyl compound (4)” or „the C-1 methyl benzoate compound (4)” instead of „the C-1 methylbenzoyl compound (4)”
Similarly, in Figure 1.: 3: 1-benzyloxy pancratistatin or pancratistatin-1-yl benzoate or 1-benzoate derivative of pancratistatin and 4: 1-benzoyloxymethyl pancratistatin or pancratistatin-1-ylmethyl benzoate or 1-methyl benzoate derivative of pancratistatin
Why are yields given in range instead of concrete values? For example, in Scheme 2: 67-74%.
L130: „Nitroso Diels-Alder reaction” instead of „Nitroso Diels-Alder”
In Scheme 3 i. and Scheme 4 iii. there are no temperature values. These took place at rt?
In Scheme 3. the yield of 12 dibromodiene diol is missing.
Unnecessary spaces and hyphens instead of em dashes: L145 (C-1), L156 (the conversion), L163 (1,4-conjugate), L209 (1-methoxycarbonyl), L224 (1-methoxycarbonyl), L334 („24 h. Once”), L412 (1,2-bis (diphenylphosphino)ethane), L425 and 443 (-Methyl)
L153 and 209: The full stop is missing from the end of the Scheme caption.
L180: The word „via” should be written in italics.
L180: vinyl instead of viny
L191 and 208: „Intramolecular Heck Reaction” instead of „Intramolecular Heck”
L191: „Global Deprotection to Afford 1-Methoxycarbonyl Narciclasine” instead of „global deprotection to afford 1-carboxymethyl narciclasine”
L230: 15.5 µM instead of 15.5 mM (micromole instead of millimole)
In general, about the experimental section:
It is somewhat irregular that elemental analysis is given for the first 3 of the 9 compounds described, but not for the others.
In the IUPAC names given, the following should be written in italics: stereodescriptors R and S, tert (for example tert-butyl), [d] (for example L270) and N (L388).
L406-408: The correct name according to Chemdraw is: (3aS,3bR,12S,12aS)-4-tert-butyl 11-methyl 12-((tert-butyldimethylsilyl)oxy)-6-methoxy-2,2-dimethyl-5-oxo-3b,5,12,12a-tetrahydrobis([1,3]dioxolo)[4,5-c:4',5'-j]phenanthridine-4,11(3aH)-dicarboxylate
The retention factor (Rf) values are usually given to 2 decimal places.
The yields given in the experimental section often do not match the values shown in the Schemes: L320 (86% comp 94%), L340 (54% comp 55%), L379 (72% comp 65%), L397 (75% comp 79%), L415 (40% comp 42%) and L433 (80% comp 79%).
Minor inaccuracies:
L294-295: THF:H2O, 10:1 while in Scheme 4 THF:H2O, 9:1
L322: 7.346 ppm rounded to 7.35 ppm
L345-347: The chemical shifts in 13C NMR assignment are usually rounded to 1 decimal place, not 2.
L379: 0 oC to rt while in Scheme 6 reflux
L409: 123 mg 28 is 0.20 mmol, not 1.27 mmol ((M=600.41g/mol)
L446: If the HCl is 1.1M, it should be eliminated from the sentence that it is conc.
L447: „then allowed” instead of „then the allowed” (unnecessary the)
L416: It is not entirely clear what 61% brsm means. Is this the yield of the recovered starting material (28, 41 mg)?
References:
[3a] and [3b]: Vshyvenko, S.; etc. instead of S. Vshyvenko, etc.
[5]: unnecessary space before the volume number
[6c], [7a], [9] and [10]: unnecessary dot after the journal name (Tetrahedron)
[6d]: the starting page number is 132505 and not 135505, and the ending page number is missing
[13k]: missing doi (crossref) (DOI: 10.1055/s-0028-1087946)
[13o]: there is an extra space in the title of the publication before the full stop
[13q]: the title of the publication is missing (Enzymatic dihydroxylation of aromatics in enantioselective synthesis: Expanding asymmetric methodology)
[13v]: the ending page number is missing (3-9)
[15]: there is an unnecessary space after the volume number
There are sometimes hyphens (for example [11]), sometimes dashes (for example [10]), or unnecessary spaces (for example [11]) between the page numbers.
Congratulations to all the authors on their successful work!
Author Response
The authors would like to sincerely thank the reviewer for their careful examination of our manuscript, and their insightful feedback. The majority of the revisions suggested by the reviewer were applied to the manuscript. Minor exceptions/comments are specified below.
Comment: 1-Hydroxynarciclasine is not described in SciFinder. Do you have a reference for it? I didn't find it at ref [6]. It would be interesting to have some information about its effect, as the target product is also designed to be functionalized in position 1 of narciclasine.
Reply: This is a derivative that has been recently made by our group and the paper describing its semi-synthesis from natural narciclasine is currently under review for submission in Molecules. This compound was also inactive when tested against the same cancer line (A549, IC50 > 500 mM). A reference was added [6d] that includes the list of authors on that paper, and that states that it has been submitted for publication.
Comment: It is somewhat irregular that elemental analysis is given for the first 3 of the 9 compounds described, but not for the others.
Reply: Although the authors agree with this statement, we do not have the samples available to obtain elemental analysis at this time.
Comment: L406-408: The correct name according to Chemdraw is: (3aS,3bR,12S,12aS)-4-tert-butyl 11-methyl 12-((tert-butyldimethylsilyl)oxy)-6-methoxy-2,2-dimethyl-5-oxo-3b,5,12,12a-tetrahydrobis([1,3]dioxolo)[4,5-c:4',5'-j]phenanthridine-4,11(3aH)-dicarboxylate
Reply: Our version of Chemdraw generates the same name as included in the manuscript: 4-(tert-butyl) 11-methyl (3aS,3bR,12S,12aS)-12-((tert-butyldimethylsilyl)oxy)-6-methoxy-2,2-dimethyl-5-oxo-3a,5,12,12a-tetrahydrobis([1,3]dioxolo)[4,5-c:4',5'-j]phenanthridine-4,11(3bH)-dicarboxylate.
Comment: The retention factor (Rf) values are usually given to 2 decimal places.
Reply: Although the authors agree with this statement, we report the Rf values to 1 decimal place due to the large amount of experimental error associated with this measurement.
Comment: L409: 123 mg 28 is 0.20 mmol, not 1.27 mmol ((M=600.41g/mol).
Reply: The manuscript states “substrate 28 (123 mg, 0.17 mmol)” and the MW of 28 is 714.7 g/mol.
Comment: L416: It is not entirely clear what 61% brsm means. Is this the yield of the recovered starting material (28, 41 mg)?
Reply: No, this is the yield of 29 based on the amount of the starting material 28 that was recovered. The overall yield of the reaction was 40% but considering that some 28 is recovered the brsm yield is also reported.
Reviewer 2 Report
The manuscript is presented in a very clear and concise manner (scientific aims and experimental work). The only negative point is the excessive self-citations (20 on 16 references). For example, are all the references in ref13 are really necessary?
You can find a few corrections/suggestions below:
page 6 line 230 micromolar
page 7, line 292 7aS
Although, not the principal point of this work: In general, it would be nice to characterize enantiomeric purity with chiral HPLC instead of giving the specific rotations.
Have you considered/tried going from compound 23 to the corresponding aldehyde by BuLi-low temp+DMF? (being one of the fastest reactions Li-halogen exchange should afford the vinyl lithiane with a good selectivity. This could be quenched by DMF subsequently)
Author Response
The authors would like to sincerely thank the reviewer for their careful examination of our manuscript, and their insightful feedback. The majority of the revisions suggested by the reviewer were applied to the manuscript. Minor exceptions/comments are specified below.
Comment: The only negative point is the excessive self-citations (20 on 16 references). For example, are all the references in ref13 are really necessary?
Reply: Reference 3c, and references 13 q through v were removed. Although the authors generally agree with this statement, the references are necessary. Many of the derivatives that are described (compounds 4-10, ref. 3 and 6) were only made by the Hudlicky group. Additionally, the approach described is heavily based on previous syntheses from the Hudlicky group – the use of dihydrodiene diols as the starting material, the Nitroso DA reaction to install the C2 and C4 heteroatoms and the intramolecular Heck reaction to form the C10a-C10b bond are all based on previous work from the Hudlicky group.
Comment: page 7, line 292 7aS.
Reply: The stereochemistry indicated (7aR) is correct.
Comment: Although not the principal point of this work: In general, it would be nice to characterize enantiomeric purity with chiral HPLC instead of giving the specific rotations.
Reply: Unfortunately, we are currently not in a position to perform such experiments.
Comment: Have you considered/tried going from compound 23 to the corresponding aldehyde by BuLi-low temp+DMF? (being one of the fastest reactions Li-halogen exchange should afford the vinyl lithiane with a good selectivity. This could be quenched by DMF subsequently)
Reply: Yes, this was attempted unsuccessfully. Deprotonation of the carbamate took place but Li – halogen exchange did not occur, even under treatment with three equivalents of BuLi base.
Reviewer 3 Report
The paper by Habaz, Kornienko and co-workers entitled “Design and synthesis of C-1 derivatives of Narciclasine and their biological activities” describes their efforts towards novel narciclasine analogs bearing comparable or improved biological activities than the natural product. To follow on from previous work reported by the same group, the authors describe herein the design, the synthesis and the biological evaluation of a new C-1 derivative (C-1 carboxymethyl analog 10). The synthesis of 10 is based on the independent preparation of 2 key fragments: 11 – bearing ring C, and 14 – bearing aromatic ring A. Intermediate 14 was prepared following a known strategy already used by the authors. The synthesis of 11 starts from o-dibromobenzene (13) and key steps include enzymatic dihydroxylation and a nitroso-Diels-Alder reaction. Then, intramolecular Heck cyclization between 11 and 14 gave advanced intermediate 29 that afforded the targeted analog 10 after its global deprotection. Concerning biological activity (in vitro cancer model-lung adenocarcinoma cell line A549), analog 10 displayed much lower inhibition compared to narciclasine (IC50 = 15.5 mM vs IC50 = 20 nM for the natural product).
Even though the work within the submitted manuscrit follows similar rational in terms of design of novel analogs of narciclasine already published by the authors (refs 6a-d), it has been well done and the manuscript is very well organized and clearly written.
The reviewer recommends thus to accept the manuscript for publication into Molecules. However, two points might be reconsidered:
- The title of the manuscript might be in singular as only one analog was synthesized and studied for its biological efficiency: “Design and synthesis of a C-1 derivative of narciclasine and its biological activity”
- In some NMR spectra (for examples: compounds 27, 30, 10), pics that do not belong to the described products need to be identified (solvent traces, grease, impurities …)
Author Response
The authors would like to sincerely thank the reviewer for their careful examination of our manuscript, and their insightful feedback. The revisions suggested by the reviewer were applied to the manuscript, with a minor exception specified below.
Comment: In some NMR spectra (for examples: compounds 27, 30, 10), pics that do not belong to the described products need to be identified (solvent traces, grease, impurities …)
Reply: The authors adjusted the spectrum of compound 10 (acetone-d6) to reflect these changes. However, the authors think that the average reader should be able to correctly identify the common impurities observed in NMR spectra obtained with chloroform-d as the solvent (i.e. water, grease, etc.).